# Flood-Related Federally Declared Disaster Events and Community Functioning (COPEWELL)

**Norma F. Kanarek** [1,*], **Qi Wang** [2], **Tak Igusa** [2], **Tara Kirk Sell** [1], **Zachary Anthony Cox** [3], **James M. Kendra** [3] and **Jonathan Links** [1]

1    Department of Environmental Health and Engineering, Bloomberg School of Public Health, Johns Hopkins University, Baltimore, MD 21218, USA
2    Department of Civil and Systems Engineering, G.W.C Whiting School of Engineering, Johns Hopkins University, Baltimore, MD 21218, USA
3    Joseph R. Biden, Jr., School of Public Policy and Administration, University of Delaware, Newark, DE 19716, USA
*    Correspondence: nkanare1@jhu.edu

**Abstract:** Objective: Understanding long-term disaster effects is key to building theories of recovery and informing policymaking. Findings regarding long-term recovery are inconsistent, with some scholars finding that disasters have little long-term impact, and others asserting otherwise. To assist in resolving this discord, we apply a conceptual framework and computational model of community resilience ("COPEWELL") that places community functioning (CF) at the center of evaluating the effects of disaster over time. Using flooding as a disaster type, we hypothesize a change in baseline CF trend when a flood-related federally declared disaster event occurs. Methods: We used county-level flood-related federally declared disaster events (2010–2014) and selected population demographics to study their effects on annual CF trends among United States counties (N = 3141). Results: In multivariate analysis of baseline CF, we found a significant negative relationship of prior five-year flood status, federal regions relative to the Northeast (Region I), lower total earnings, and greater population size. Annual CF trend was 0.09% (95%CI: 0.01%–0.16%). In multivariate analysis, significant predictors included baseline CF (β = −0.0178, −0.0047−−0.0309), any concurrent flood-related federally declared disaster events (−0.0024, −0.0040−−0.0008), ten-year prior flood events (−0.0017, −0.0034−−0.0000) and concurrent population change (−0.0186, −0.0338−−0.0035). Conclusions: Recent floods depress baseline CF, while concurrent and ten-year-ago floods depress trend in CF. Resilience may potentially be modified by raising baseline CF and maintaining population over time.

**Keywords:** disaster; flood; hurricanes; community functioning; recession; population change

## 1. Introduction

Understanding the effects of disasters of all kinds is key to enhancing both resistance to and recovery from significant events. Moreover, as disasters proliferate, we need improved ways of understanding the effects of multiple sequential events. Research and policy interventions have largely construed disasters as isolated events. Nevertheless, the experience of the last 15 years or so shows the precariousness of some locations, and the possibility of experiencing disastrous events again and again. Some places may, in fact, endure a kind of continuous disaster, never fully recovering or recovering only partially prior to the impact of another hazard. For example, New Orleans held its breath as Hurricane Rita missed them just a few weeks after Hurricane Katrina. In 2004, three hurricanes crisscrossed Florida 12 years after Hurricane Andrew. A year later Hurricane Wilma swept over a similar area. More recently the last few years have seen multiple flooding events in the same parts of North Carolina. Simultaneous with shifts in environmental hazards and human occupancy of hazardous areas (and their iterative effects) is the precariousness

in the economy, which may shift the boundaries of vulnerability, making people more vulnerable over time. The last 20 years have seen the dot-com crash, the housing crash and, most recently, the COVID crash. Scholarly and policymaking interest in recovery has increased, as well, over these timeframes.

In addition, the overall scholarly literature on disaster recovery is in conflict. Studies have found that "Pre-disaster trends (e.g., of economic growth or decline) are often accelerated, exacerbated, or intensified in recovery," but at the same time, some studies have found little long term-impact and other works have indeed found long-term changes, especially after catastrophic events affecting the economy [1]. Thus a current research need is to help show which of the preceding possible tendencies is more prevalent.

There is no consensus on the accepted measures of disaster recovery that might be used in benchmarking recovery in the required data analysis. Typically, such measures might include housing, infrastructure restoration, business activity, and provision of health services [2–5]. Our current conceptual framework and computational model for the predicted time-course of event effects, COPEWELL, considers community functioning as the concept of interest [6].

Full details of the development of the COPEWELL model and the development and validation of the measures are available elsewhere [1]. The COPEWELL system dynamics model [6] frames community functioning (CF) as a feature of resilience whose gain or loss after an event is dependent on many factors, including pre-event CF; the size and nature of the event; population vulnerability, inequality and deprivation; social cohesion; preparedness and response; external resources; natural systems; engineered systems; and countermeasures [6]. Furthermore, the COPEWELL model predicts a drop in community functioning following an event such as a federally declared disaster, accompanied afterwards by a bounce back toward initial CF levels over time. This means that repeated events in a short period of time may compound each other and work to lower baseline CF for the subsequent event before it recovers completely from the preceding event [7]. Of note, the time-course of CF during and after an event allows the calculation of resilience (also measured by COPEWELL) [8].

In COPEWELL, CF is multifaceted and is composed of 27 measures of communication, education, food and water, government, housing, health care and public health, nurturing and care, transportation, and well-being, based on publicly-available for United States (US) counties. These measures may be subject to variations caused by natural events such as flood or human-caused events such as an economic downturn, which in turn would influence COPEWELL's predictions of resilience.

As an example of a human-caused event, the Great Recession of 2007–2009 had far-reaching effects on a number of societal sectors [9]. There is evidence that economic downturns negatively affect healthy dietary choices [10], population trends in healthier alcohol consumption patterns [11], early detection of disease [12], employment [9,13], local public health spending [14], health equity (through government spending reductions and challenges in the labor market) [15], fewer anthropogenic constituents of ambient air particulate matter [16], corporate leadership [17], lowering return on investment in services for the elderly [18] and per capita government spending [9]. A worsening or improving national economy may also affect local jurisdictions. Literature to date about flood events shows changes in depression medication prescribing [19], illness induction and worsening population vulnerability [20], lower birthweights [21], pre-event "vulnerability" [22], worsening health [23], and infrastructure disruption [24].

Using the COPEWELL framework and data from several societal sectors, we seek to validate changes in CF from flood-related federally declared disaster event(s) during a period (2010–2015) that can be characterized as post-recession (2007–2009) [9]. We evaluated (1) whether there is a positive trend in CF; (2) whether experiencing a flood-related federally declared disaster event during the same time period is an independent negative predictor; (3) whether baseline CF (2010) is related positively to CF trend post-recession;

and (4) whether the occurrence of floods in prior periods is an influence on CF trend more recently.

## 2. Methods

### 2.1. Data Sources

#### 2.1.1. Floods

Federally declared disaster events are a combination of state governor request (on behalf of part or all of the state) for emergency funds and a presidential declaration of disaster. (https://www.fema.gov/disaster-recovery-reform-act-2018) (accessed on 1 January 2017) FEMA captures these events, which were found in the online file maintained by the US Office of Preparedness (https://www.fema.gov/openfema-dataset-disaster-declarations-summaries-v1) (accessed on 1 January 2017). Event number, state and county FIPs codes, event begin and end date, event type and description are tabulated. If "flood" or "flooding" was not mentioned in the event description, the events were excluded from this analysis. Events occurring between 1995 and 2014 were grouped in 5 year intervals: 1995–1999, 2000–2004, 2005–2009, and 2011–2014. 2011–2014 is referred as the study period; 2005–2009 as the prior 5 year period; 2000–2004 as the prior 10 year period; and 1995–1999 as the prior 15 year period.

#### 2.1.2. Community Functioning

CF was a calculated combination conveyed as an index having values between 0 and 1. Constituent measures were described previously by Links et al. [6] CF for years subsequent to 2010 was indexed to 2010 means and standard deviations.

#### 2.1.3. County Characteristics

Additional county characteristics were downloaded from various online data sources, such as the US Census [18], the American Community Survey [19], and the Roadmap to Health database [20].

### 2.2. Dependent Variable

Average annual percentage (%) change, CF trend, was calculated using annual CF from 2010 to 2015 for each county so that 5 year to year changes were available for calculating CF trend.

### 2.3. Independent Variable

Flooding is a common occurrence in events leading to a disaster declaration. Amount (number), intensity (type of FDD event) and duration (length of flood-related events during the period) may be characterized.

#### 2.3.1. County with a Flood

A county was deemed as being flood affected when at least one flood declaration was made during the study period. Values included yes (1) and no (0). Flood yes or no was ascertained for 2011–2014 (study period) and previous time periods of 2005–2009 (prior five year period), 2000–2004 (prior ten year period), and 1995–1999 (prior fifteen year period).

#### 2.3.2. Flood Duration

Length (days) of a flood-related event was the difference between the declaration event end date and begin date. Because some events began and ended on the same calendar day, half a day was added to all flood durations for any county with a flood event. Counties having no study period flood event were coded as zero in length.

### 2.3.3. Flood Type

Event type was coded by FEMA and re-categorized in this study as coastal storm or tsunami, flood, hurricane or typhoon, and severe storms or tornados. Events such as storms that had no mention of flood or flooding were excluded.

### 2.3.4. Number of Flood-Related Events

Number of flood-related events was a count of events enumerated by county. Cumulative county count ranged from 0 to 9 events.

### 2.3.5. Year of Event

Year of event was extracted from the federally declared start date and had year values ranging from 1995 to 2014. Counties could have flood-related federally qualified disaster events in more than one year and experience more than one event per year. In the FEMA database county-event combination was unique, aggregated over the time period and was represented only once in our analysis database.

### *2.4. Covariates and Potential Confounders*

### 2.4.1. Population Size

2010 population counts were continuous. Population size was also categorized as less than 50,000, 50,000–99,999 and 100,000 or more population.

### 2.4.2. Population Density

Population density (2010) was defined as persons per square mile. Density was categorized by quartiles (Quartile 1 = 0–16.8, Quartile 2 = 16.9–45.1, Quartile 3 = 45.2–113.3, Quartile 4 = 113.6–68,951 persons per square mile).

### 2.4.3. Population Change

Population change (%) as a continuous measure was calculated for the period 2010–2015. Positive change means population growth above the baseline year. We also categorized population change by tertiles (Tertile 1: less than −0.017%, Tertile 2: −0.017–0.0136%, and Tertile 3: greater than 0.0136%).

### 2.4.4. Total Earnings

Total earned income for 2010 was continuous and is presented in billions of dollars. Total earned income also was dichotomized as below or above $1 billion dollars.

### 2.4.5. United States Regions

Federal Emergency Management Agency (FEMA) region numbers were assigned to county geographical locations. (https://www.fema.gov/fema-regional-contacts) (accessed on 1 January 2017) This categorization was made to ten regions.

### 2.4.6. Closest Coast

States were coded as being closest to the Atlantic, Gulf, or Pacific Coast as determined by ordinal tiers from the coast. States abutting one of these major bodies of water were coded as closest, states next to these states were coded as next closest, and so on. When there was a tie, distance was used to break the tie. Values were ordinal.

### *2.5. Institutional Review Board*

This study was not human subjects research, so IRB approval was not sought.

### *2.6. Units of Analysis*

Communities numbered 3141 unique US counties or county aggregates for which there were CF data for 2010 to 2015.

Statistical analysis was conducted using Stata 11. Chi square or ANOVA provided statistical tests of differences between groups. Multivariable modeling was conducted as ordinary regression. A forward model building approach was used.

## 3. Results

### 3.1. Characteristics of Flood-Related Events

During the study period, 3560 unique county-events were recorded as flood-related FDD events. The majority (44.0%) were flood-related severe storms or tornadoes. Next in frequency were hurricane or typhoon (27.7%), flood (23.3%), and coastal storm or tsunami (5.0%). More than half of county-events occurred in Regions VII (19.4%), IV (17.1%), and III (15.8%). The majority of severe storms or tornadoes occurred in three Regions: II (23.6%), III (17.7%), and VI (17.2%). A third of hurricanes or typhoons occurred in Region VII (32.5%) and a quarter occurred in Region III (27.1%), similarly about a third of floods occurred in Region VII counties (34.1%) with a quarter more in each, Region IV and Region V. Coastal storms or tsunamis predominated just in region IV (92.7%). (Table 1).

**Table 1.** Distribution of FEMA event type by year, FEMA region, coast region, tier from major coast, length of disaster and average county population size, by county, 2011–2014.

| | **FEMA Event Type** | | | | |
|---|---|---|---|---|---|
| | **All FEMA Flood-Related Declarations** | **Coastal Storm, Tsunami** | **Flood** | **Hurricane, Typhoon** | **Severe Storms, Tornados** |
| All Events (%) | 3560 (100%) | 179 (5.0%) | 829 (23.3%) | 987 (27.7%) | 1565 (44.0%) |
| Year (%) *** | | | | | |
| 2011 | 1687 (47.4) | 140 (78.2%) | 263 (31.7%) | 521 (52.8%) | 763 (48.8%) |
| 2012 | 810 (22.8) | 18 (10.1) | 92 (11.1) | 263 (26.6) | 437 (27.9) |
| 2013 | 540 (15.2) | 0 (0.0) | 237 (28.6) | 127 (12.9) | 176 (11.3) |
| 2014 | 523 (14.7) | 21 (11.7) | 237 (28.6) | 76 (7.7) | 189 (12.1) |
| DHHS Region (%) *** | | | | | |
| I | 214 (6.0%) | 0 (0.0%) | 68 (8.2%) | 97 (9.8%) | 49 (3.1%) |
| II | 370 (10.4) | 0 (0.0) | 0 (0.0) | 0 (0.0) | 370 (23.6) |
| III | 561 (15.8) | 0 (0.0) | 17 (2.1) | 267 (27.1) | 277 (17.7) |
| IV | 608 (17.1)) | 166 (92.7) | 228 (27.5) | 0 (0.0) | 214 (13.7) |
| V | 514 (14.4) | 0 (0.0) | 220 (26.5) | 144 (14.6) | 150 (9.6) |
| VI | 425 (11.9) | 0 (0.0) | 0 (0.0) | 156 (15.8) | 269 (17.2) |
| VII | 692 (19.4) | 0 (0.0) | 283 (34.1) | 321 (32.5) | 88 (5.6) |
| VIII | 128 (3.6) | 8 (4.5) | 0 (0.0) | 2 (0.2) | 118 (7.5) |
| IX | 13 (0.4) | 0 (0.0) | 8 (1.0) | 0 (0.0) | 5 (0.3) |
| X | 35 (1.0) | 5 (2.8) | 5 (0.6) | 0 (0.0) | 25 (1.6) |
| Coast Region (%) *** | | | | | |
| Atlantic | 1452 (40.8%) | 0 (0.0%) | 325 (39.2%) | 213 (21.6) | 914 (58.4) |
| Gulf | 1784 (50.1) | 166 (92.7) | 431 (52.0) | 772 (78.2) | 415 (26.5) |
| Atlantic/Gulf | 60 (1.7) | 0 (0.0) | 13 (1.6) | 2 (0.2) | 0 (0.0) |
| Pacific | 264 (7.4) | 13 (7.3) | 60 (7.2) | 0 (0.0) | 236 (15.1) |
| Tier from Major Coast (%) *** | | | | | |
| 1 | 1735 (48.7%) | 171 (95.5) | 321 (38.7) | 520 (52.7%) | 723 (46.2%) |
| 2 | 491 (13.8) | 0 (0.0) | 5 (0.6) | 0 (0.0) | 486 (31.1) |
| 3 | 618 (17.4) | 8 (4.5) | 118 (14.2) | 342 (34.7) | 150 (9.6) |
| 4 | 591 (16.6) | 0 (0.0) | 385 (46.4) | 0 (0.0) | 206 (13.2) |
| 5 | 125 (3.5) | 0 (0.0) | 0 (0.0) | 125 (12.7) | 0 (0.0) |

**Table 1.** *Cont.*

| | FEMA Event Type | | | | |
|---|---|---|---|---|---|
| | **All FEMA Flood-Related Declarations** | **Coastal Storm, Tsunami** | **Flood** | **Hurricane, Typhoon** | **Severe Storms, Tornados** |
| Average length of disaster *** (SD) | 18.4 (24.8) | 33.1 (14.7) | 14.1 (16.7) | 20.2 (21.3) | 17.7 (30.1) |
| Average Population Size 2010 *** (SD) | 124,279 (318,843) | 76,865 (117,473) | 118,370 (325,147)) | 115,669 (234,761) | 183,135 (309,703) |
| Average population Density (SD) * | 439 (2726) | 404 (1383) | 531 (3158) | 1004 (5098) | 1146 (4898) |
| Average County Square Miles (SD) ** | 978 (2885) | 712 (646) | 791 (812) | 883 (1102) | 700 (474) |
| Average Total Earnings (billions) (SD) ns | $2.6 (9.9) | $3.6 (9.1) | $2.7 (9.6) | $3.9 (14) | $3.3 (6.4) |

*: $p$ value < 0.001; **: $p$ value < 0.01; ***: $p$ value < 0.05; and ns: statistically not significant.

Of all flood-related events half were in jurisdictions closest to the Gulf of Mexico (50.1–51.8%) and next, those nearest the Atlantic Ocean (40.8–42.5%) (Table 1).

Coastal storms had the longest event lengths and affected counties of the smallest average population size. Floods had the shortest event length. Hurricane- or typhoon-affected counties had the largest average county areal size and the highest total earnings. Severe storms or tornado-affected counties were largest in population size (Table 1).

*3.2. County Characteristics by Flood Status*

Table 2 presents counties and their characteristics by flood status. The largest proportion of counties incurring flood-related events was found in 2011 (987/3141 = 31.4%). The number of counties affected decreased from 2011 to 2014. In all, 51.0% (1601/3141) of counties encountered a flood-related event. Counties in FEMA Regions I, II, III, and VI had the largest average number of events per county. Counties that experienced a flood were also quite likely (1019/1601 = 63.6%) to have more than one flood-related event between 2010 and 2015 (data not shown). Counties in states which were respectively closest to the Atlantic Ocean and/or the Gulf of Mexico have the most per-county events. The average county length of all events, in a year with events, was 40.7 days. Individual events averaged about half that (19.3 days). Though population size and total earnings were similar, population density tended to be higher among counties having floods 2011–2014 (Table 2).

**Table 2.** FEMA Event and No-FEMA Event Counties and Events Rates, 2011–2014.

| | Unique Counties (N = 3141) | | | |
|---|---|---|---|---|
| | **Number of Counties** | | **Mean** | |
| | **No FEMA Flood-Related Events** | **Any FEMA Flood-Related Events** | **Events Per FEMA Flood-Related Event Counties (N = 1601)** | **Events Per All Counties (N = 3141)** |
| Counties | 1540 (49%) | 1601 (51%) | 2.22 (2.15–2.29) | 1.13 (1.09–1.17) |
| Year | | | | |
| 2011 | 2154 (69%) | 987 (31%) | 3.10 (2.12–4.08) | 1.71 (1.65–1.77) |
| 2012 | 2556 (81%) | 585 (19%) | 1.20 (0.90–1.50) | 1.38 (1.34–1.43) |
| 2013 | 2675 (85%) | 466 (15%) | 1.10 (0.87–1.33) | 1.16 (1.12–1.19) |
| 2014 | 2701 (86%) | 440 (14%) | 1.10 (0.87–1.33) | 1.19 (1.15–1.23) |

**Table 2.** *Cont.*

| | Unique Counties (N = 3141) | | | |
| --- | --- | --- | --- | --- |
| | Number of Counties | | Mean | |
| | No FEMA Flood-Related Events | Any FEMA Flood-Related Events | Events Per FEMA Flood-Related Event Counties (N = 1601) | Events Per All Counties (N = 3141) |
| DHHS Region | $p = 0.0000$ | | $p = 0.0000$ | $p = 0.0000$ |
| I | 19 (1.2%) | 48 (3.0%) | 4.96 (4.25–5.67) | 3.55 (2.81–4.29) |
| II | 0 (0.0%) | 83 (5.2%) | 4.17 (3.81–4.53) | 4.17 (3.81–4.53) |
| III | 140 (9.1%) | 142 (8.9%) | 2.75 (2.48–3.03) | 1.39 (1.17–1.60) |
| IV | 328 (21.3%) | 408 (25.5%) | 1.91 (1.82–1.99) | 1.06 (0.97–1.14) |
| V | 189 (12.3%) | 335 (20.9%) | 1.53 (1.46–1.61) | 0.98 (0.90–1.06) |
| VI | 342 (22.2%) | 161 (10.1%) | 2.64 (2.41–2.87) | 0.84 (0.72–0.97) |
| VII | 87 (5.6%) | 325 (20.3%) | 2.13 (2.01–2.24) | 1.68 (1.56–1.80) |
| VIII | 194 (12.6%) | 68 (4.2%) | 1.88 (1.62–2.15) | 0.49 (0.37–0.61) |
| IX | 117 (7.6%) | 7 (0.4%) | 1.86 (1.19–2.52) | 0.10 (0.02–0.19) |
| X | 124 (8.1%) | 24 (1.5%) | 1.46 (1.17–1.75) | 0.24 (0.14–0.33) |
| Coastal Region | | | | |
| Atlantic | | 584 (36.5% | 2.49 (2.34–2.63) | |
| Gulf | | 817 (51.0%) | 2.18 (2.11–2.26) | |
| Atlantic/Gulf | | 47 (2.9%) | 1.28 (1.11–1.45) | |
| Pacific | | 153 (9.6%) | 1.73 (1.57–1.88) | |
| Events Length | | | | |
| Each events | | | 19.3 (18.1–20.4) | |
| All events | | | 40.7 (38.4–43.0) | |
| Average Population Size 2010 *** | $p = 0.6167$ | | | |
| | 101,017 (88,983–113,052) | 95,424 (76,920–113,928) | | |
| Average Population Density 2010 (per sq mile) * | $p = 0.0445$ | | | |
| | 196 (159–233) | 319 (207–431) | | |
| Total Earnings ($million) | $p = 0.4168$ | | | |
| | 1730 (1310–2150) | 1970 (1570–2370) | | |

*: *p* value < 0.001; ***: *p* value < 0.05.

### 3.3. County Characteristics by Measures of Community Functioning

Baseline CF (2010) and percent change CF (2010–2015) differed by FEMA region, population size, population density, and total earnings. (Table 3) These three measures differed in pattern of effect between CF baseline and CF trend. For example, the largest population size counties were found to have the highest baseline CF, measuring above the index median of 0.500, while with regard to CF trend, the largest counties displayed a significant decline in CF. On the other hand, counties of small size, under 50,000, have significant and positive CF trends. Population density was observed to have a u-shaped distribution for CF 2010 yet for CF trend, population density's positive relationship and significance was confined to the lowest population density category. CF 2010 was significantly above average for counties with annual total earnings less than $1 billion yet the trend in CF exhibited a negative trend for these same counties (Table 3).

CF 2010 was statistically and significantly above the index average CF of 0.500 in FEMA Regions I, II, V, VII, VIII and X (Table 3). With regard to trend, all counties had an upward trend of 0.09% per year and FEMA Regions V and X were alone in having a statistically increasing trend for 2010–2015 (Table 3).

**Table 3.** Mean baseline CF (2010) and CF Trend (2010–2015), all US Counties.

| | Mean (CI) CF$_{2010}$ | Mean (CI) Trend CF$_{trend}$ (%) |
|---|---|---|
| Overall | 0.500 (0.498–0.502) | 0.09% (0.01–0.16) |
| DHHS Region | *p* < 0.0000 | *p* = 0.0303 |
| I | 0.560 (0.553–0.567) | 0.14% (−0.06–0.34%) |
| II | 0.529 (0.521–0.537) | 0.03 (−0.21–0.28) |
| III | 0.499 (0.492–505) | 0.02 (−0.18–0.21) |
| IV | 0.451 (0.448–0.454) | −0.14 (−0.30–0.02) |
| V | 0.504 (0.501–0.508) | 0.26 (0.15–0.38) |
| VI | 0.483 (0.479–0.486) | 0.05 (−0.17–0.27) |
| VII | 0.544 (0.540–0.549) | 0.17 (−0.03–0.36) |
| VIII | 0.568 (0.561–0.576) | 0.10 (−0.29–0.48) |
| IX | 0.493 (0.484–0.502) | 0.19 (−0.23–0.60) |
| X | 0.512 (0.504–0.520) | 0.53 (0.04–1.02) |
| Average Population Size 2010 *** | *p* < 0.0000 | *p* = 0.0000 |
| <50,000 | 0.501 (0.498–0.504) | 0.15 % (0.04–0.26) |
| 50,000–99,999 | 0.487(0.482–0.491) | −0.02 (−0.16–0.11) |
| 100,000+ | 0.507 (0.504–0.511) | −0.08 (−0.15–−0.01) |
| Average Population Density(Quartiles) | *p* < 0.0000 | *p* = 0.0000 |
| 0 | 0.5376 (0.5328–0.5423) | 0.30% (0.06–0.54) |
| 1 | 0.4833 (0.4794–0.4871) | 0.09 (−0.06–0.23) |
| 2 | 0.4760 (0.4729–0.4791) | 0.01 (−0.10–0.11) |
| 4 | 0.5040 (0.5007–0.5073) | −0.05 (−0.13–0.03) |
| Total Earnings ($million) | *p* < 0.0000 | *p* = 0.0000 |
| <1000 | 0.5105 (0.5072–0.5138) | −0.09% (−0.16–−0.02) |
| ≥1000 | 0.4973 (0.4948–0.4998) | 0.14 (0.04–0.23) |
| Population change (Tertiles) | | *p* = 0.0000 |
| T1 | | 0.14% (−0.03–0.29) |
| T2 | | 0.12 (0.00–0.25) |
| T3 | | 0.01 (−0.11–0.12) |
| Flood Status | | |
| 2010–2014 | | *p* = 0.0009 |
| No | — | 0.22% (0.09–0.35) |
| Yes | — | −0.04 (−0.13–0.05) |
| 2005–2009 | *p* = 0.0000 | *p* = 0.0000 |
| No | 0.5454 (0.5327–0.5580) | 0.43% (−0.30–1.15) |
| Yes | 0.4990 (0.4969–0.5011) | 0.08 (0.00–0.15) |
| 2000–2004 | *p* = 0.0000 | *p* = 0.0000 |
| No | 0.5121 (0.5085–0.5156) | 0.25% (0.10–0.41) |
| Yes | 0.4937 (0.4912–0.4962) | −0.00 (−0.09–0.08) |
| 1995–1999 | *p* = 0.0000 | *p* = 0.0000 |
| No | 0.5075 (0.5038–0.5111) | 0.15% (0.00–0.29) |
| Yes | 0.4965 (0.4940–0.4990) | 0.06 (−0.03–0.15) |

***: *p* value < 0.05.

### 3.4. Measures of Flood

Table 4 shows the effect of flood(s) duration, frequency, and ever adjusted for baseline CF and population change. Each was confirmed as an independent predictor of flood effects on trend.

Our findings show counties with flood-related events had declines in CF trend; whereas counties without flood-related events had positive CF trends (0.22%, 95% CI: 0.09–0.35%); and the unadjusted odds ratio showed a reduction in trend when there were flood-related event(s) relative to no flood events (0.832, (95%CI: 0.7286 to 0.9642)) (data not shown).

**Table 4.** Multivariate models of CF trend (2010–2015) in counties with a FEMA-related flood event.

| | Flood Duration Model (Beta, CI) N = 3139 | Flood Frequency Model (Beta, CI) N = 3139 | Flood Ever Model with Population Change |
|---|---|---|---|
| Flood Status 2010–2014 Yes/No | | | −0.0028 *** (−0.0043−−0.0013) |
| Number of flood events | | −0.0009 *** (−0.0014−−0.0003) | |
| Length of events | −0.00004 *** (−0.00006−−0.00002) | | |
| Community Functioning 2010 | 0.0274 *** (0.0122–0.0427) | 0.0210 *** (0.0080−−0.0339) | 0.0199 *** (0.0069–0.0328) |
| Population Change 2010–2015 | −0.0128 ns (−0.0304–0.0047) | −0.0170 ** (−0.0321−−0.0020) | −0.0179 ** (−0.0330−−0.028) |

***: $p < 0.01$; **: $p < 0.05$; ns: statististically not significant.

### 3.5. Fully Adjusted Models of CF Trend

Multivariable modeling results are shown in Table 5, which displays the fully adjusted models for baseline CF and CF trend. The final models illustrated that county characteristics (FEMA region, total earnings, population size) are significant in modelling baseline CF. As well, the most recent, prior five year period flood (2005–2009) was a significant factor predicting baseline CF. On the other hand, CF trend is predicted by by baseline CF, population change, and prior ten year period floods (2000–2004) alone.

**Table 5.** Multivariate model of Baseline CF (2010) and CF Trend (2010–2015).

| | Baseline CF (Best Model) (Beta, CI) | CF Trend (Best Model) (Beta, CI) |
|---|---|---|
| Flood Status Y/N | | |
| 2010–2014 | — | −0.0024 *** (−0.0040−−0.0008) |
| 2005–2009 | −0.0250 *** (−0.0356−−0.0144) | — |
| 2000–2004 | — | −0.0017 * −0.0034−−0.0000) |
| 1995–1999 | — | — |
| Community Functioning 2010 | | 0.0178 *** (0.0047–0.0309) |
| EPA Region | | |
| I | Reference | |
| II | −0.0349 *** (−0.0499−−0.0198) | |
| III | −0.0626 *** (−0.0749–0.0503) | |
| IV | −0.1061 *** (−0.1176–0.0945) | |
| V | −0.0532 *** (−0.0649–0.0415) | |
| VI | −0.0744 *** (−0.0862−−0.0627) | |
| VII | −0.0127 * (−0.0246–0.0008) | |
| VIII | 0.0093 ns (−0.0031–0.0217) | |
| IX | −0.0670 *** (−0.0807−−0.0532) | |
| X | −0.0489 *** (−0.0622−−0.0354) | |
| Total Earnings | $1.34 \times 10^{-12}$ *** ($8.55 \times 10^{-13}$–$1.83 \times 10^{-12}$) | |
| Population | $-1.94 \times 10^{-8}$ *** ($-3.08 \times 10^{-8}$−$-8.07 \times 10^{-9}$) | |
| Population Density | $8.62 \times 10^{-7}$ ns ($-4.05 \times 10^{-7}$–$2.13 \times 10^{-6}$) | |
| Population Change | | −0.0186 ** (−0.0338−−0.0035) |

***: $p < 0.01$; **: $p < 0.05$; *: $p < 0.10$; ns: statististically not significant.

## 4. Discussion

To our knowledge, this study is the first to assess flood effects on CF among all US counties. Our outcome, the annual CF trend calculated by the COPEWELL model [6], was a multi-year measure not found in the literature previously. CF trend was used to evaluate flood effects in the context of positive change of CF. Concurrent floods and floods ten years prior indicate independent negative effects on CF trend after adjustment for population change and baseline CF.

### 4.1. Flood County-Events

The unadjusted CF trend observed in the study period was positive, overall at 0.09% (0.01–0.16%). This trend has confirmed that the increase observed was during a time of post-recession economic recovery; it was observed in small, low density, and high earnings counties. In the best multivariate model (adjusted for flood event during 2010–2014, flood event during 2000–2004, baseline CF (2010), and population change (2010–2015)), the average annual CF trend was 0.48% (0.3–1.1%).

Floods are a common FEMA disaster occurrence, with flood events affecting just over half of US counties between 2010 and 2014, the study period. During this period the average was one and a half flood events among flood affected counties. Flood events differed in nature and geographical distribution, as indicated by the federally declared disaster event type. Severe storms or tornadoes touched counties in every FEMA region of the country though they were most often near the Atlantic Coast.

### 4.2. Baseline CF

CF 2010 was considered the baseline year. Contemporaneous smaller population, more income and several FEMA regions presented with average baseline CF values of 0.5000 or above. Having no flood for any of the prior periods was also associated with having a baseline CF significantly greater than 0.5000. Furthermore, having experienced a near-term flood—2005–2009, diminished baseline CF in 2010. This confirms that prior flood events have a negative correlation with baseline CF. The COPEWELL model's prediction that CF prior to an event is positively correlated with subsequent CF trends.

With respect to baseline CF, population size, total earnings, and FEMA region underpin the actual baseline CF value, as indicated by the multivariate model of baseline CF. The effect of an (point in time) event is felt both at the moment of occurrence and over time, as a factor in longer-term recovery. In other words, county characteristics are important and act through baseline CF to influence CF trend.

### 4.3. CF Trend and Flood

Counties having had a flood during 2005–2009 had a declining 2010–2015 CF trend. We estimate this translates to an additional 70 counties having a declining CF due to flood events.

Several characteristics of a flood, namely event type, frequency and duration, adjusted for baseline CF, were shown to be negatively associated with trend in CF, confirming the robustness of flood measures in predicting CF trend.

Bivariate relationships with CF trend were plentiful. Positive trends were, for instance, linked to being in FEMA Region V or X, having small population size and density, greater earnings, modest fluctuations in population, and previously having no floods. Despite these correlations, forward model building yielded a simple model of study period flood and prior 10 year period flood (2000–2004) and population change (borderline significance). We uncovered a negative relationship with CF trend that was independent of baseline CF. This suggests that prior experience doesn't ameliorate the effects of a past flood and its effect as observed in the baseline CF. That is, while there may be some benefits to disaster experience, on the whole, at least regarding community functioning, prior disasters have a negative effect. Perhaps not surprisingly, communities that begin with greater community functioning show greater community functioning after the flood. For this reason, while not typically thought of as "mitigation," attention to and investment in the domains of community functioning are important for disaster management.

In addition, different timing of prior floods affected both baseline CF and CF trend, leading us to assume that in counties with prior floods, strengthening baseline CF may be essential to increasing resilience. Assuring and assisting individuals to remain or to return to the area after a flood will strengthen CF trend and enhance recovery [8].

Our findings in general are that floods do matter; disasters do matter in the overall trajectory of community functioning. We have already suggested an implication of this

finding: that bolstering community functioning means better community functioning after disaster. Healthier, more robust communities with greater levels of well-being across several indicators, do better on the whole. But there's more: not having any floods is also better. Their effects are not easily erased, even over time. This means that flood management efforts are good for the community. This finding, from an independent theoretical and methodological direction, supports conclusions drawn by other research finding a positive benefit: cost ratio to disaster mitigation [25]. While we make no benefit: cost assertion (which Godschalk et al. [25] noted can vary widely across hazard types), the benefits of not having floods are clear, because their impact saps a community's wellbeing over time.

### 4.4. Potential Study Limitations

This study has potential limitations in that COPEWELL data for each county was subject to statistical variation, especially in the smallest jurisdictions. Additional variation may have been introduced by using period measures of flood. Multiple flood measures were examined; though there were differences in magnitude and frequency, the direction of effect was constant; i.e., presence of flood reduced CF trend. Barr and Taylor-Robinson [9] demonstrated that CF fluctuates over time with different patterns. Pursuit of these distributional and lag-time differences might be incorporated in future analyses as we better understand performance of measures in multiple counties prior to and after an event.

There is little known about the completeness and accuracy of the FEMA dataset. For instance, in this study of counties, a few flood-related federally declared disaster events were reported as state-wide events rather than county-specific and in some cases the geographic area was smaller than a county or county aggregate. Another desirable measure of flood effects would CF change by each event, as for example CF change measured for Hurricane Sandy affected counties. Because presidential disaster declarations happen on a state by state basis, the same disaster may be described differently by state. For example, the extent of Super Storm Sandy would require additional coding by FEMA to produce an aggregation of Sandy-related effects. This and perhaps other measures would allow analytic control for the size of a particular event and thus an event's unique properties.

At least one other report [26] has suggested that migration before and during a recession is small. Our finding of a negative effect in the context of recession recovery is suggestive of population change due to other factors—perhaps the occurrence of a flood or climate-change-induced sea-level rise [27]. Additional studies during varied economic times, adjusted for sea-level rise could shed light on the durability of this correlation.

In summary, we were able to answer our original questions: (1) there is a positive trend in CF from 2010–2015, post Great Recession, (2) having a flood-related federally declared disaster event following the Great Recession is an independent negative predictor of CF trend, (3) baseline CF is related positively to CF trend, and (4) the occurrence of floods in periods prior to 2010 is a negative predictor of CF trend.

### 5. Conclusions

In this examination of all United States counties, we were able to confirm that CF has a positive trend during the study period. Prior and current flooding history are independent dampers on a county's ability to bounce back.

Conceptually, baseline CF is highly intertwined with county characteristics, including prior 5 year floods, and must be measured in any study of trend in county functioning. Improvement of CF prior to another event will enhance resilience; when there are limited prospects of improving pre-event CF, resilience could be enhanced by improvements in other COPEWELL model factors such as strengthened social cohesion or the built environment.

**Author Contributions:** Conceptualization, N.F.K. and J.L.; methodology, N.F.K., Q.W., T.I. and N.F.K.; formal analysis, N.F.K.; data curation, Q.W. and N.F.K.; writing—original draft preparation, N.F.K.; writing—review and editing, T.K.S., T.I., Z.A.C., J.M.K. and J.L.; project administration, N.F.K.; funding acquisition, J.L. and J.M.K. All authors have read and agreed to the published version of the manuscript.

**Funding:** This work was funded in part by the Center for Disease Control and Prevention, grant number 200-2017-94105.

**Institutional Review Board Statement:** This work is not human subjects research, so IRB approval was not sought.

**Informed Consent Statement:** Not applicable.

**Data Availability Statement:** No new data were created or analyzed in this study. Data sharing is not applicable to this article.

**Acknowledgments:** Valerie Marlowe and Nuno Martins (Biden Center for Public Policy) read the final manuscript.

**Conflicts of Interest:** The authors declare no conflict of interest.

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
