# Peer review of "Flood-Related Federally Declared Disaster Events and Community Functioning (COPEWELL)"

_climate, doi:10.3390/cli10110159_

Round 1

Reviewer 1 Report

I have carefully read the book proposal by Kanarek et al. and find that it is suitable for publication after minor, yet extensive, revisions. This is a straightforward, though not minor, study. The authors apply a pre-existing model and conceptual framework (COPEWELL) linking an operationalization of societal well-being (“Community Functioning” or CF) to flood-related disasters. The questions of the study are simple and clearly outlined in the introduction. As the authors explain, these questions are not trivial given the mixed responses from the state-of-the-art literature which alternatively places some role or no role in community functioning of flooding events.

The authors answer these questions clearly in the conclusion section and in a more nuanced manner in the discussion section. In particular, the authors are attentive to some of the possible limitations of the dataset and the representativity. My comments and calls for revision are mostly connected to requests for clarification and apparent typos. My comments follow below.

Comments

-The authors often use the term “precariousness”. Is this a deliberate decision not to use the term “vulnerability” which is used and defined in the IPCC? Is “precariousness” different? from “vulnerability”? Is the point irrelevant to this study?

-(Intro, 1st Par, bottom line) Is it helpful to conflate economic disturbances with environmental disturbances? Are the authors suggesting that recovery from these events would be similar? I ask because, in the following paragraph, the only reference given is specific to economic disasters.

-(Intro, 5th  Par) word “data” is missing, methinks.

-(2.1.2) Is “2010” an error?

-3.1 How are “typhoons” included here? Are these from the Pacific territories? Why would we include these? They are completely different than US-based communities. What is the percentage of Typhoon/(Typhoon + Hurricanes)?

-Discussion section:

--Could the authors please define “current” as in “current floods” and “floods ten years prior”? I understand that the most recent data is from 2015.

--4.1, 1st Par, Last sentence. The sentence is not very clear to me. Unless I am mistaken the “)” should be moved back to include “and population change (2010-15)”. Either that or the sentence should be rewritten.

--4.1 2nd Par. The statement “counties tended to have more than one flood event, as the average was one and a half events each for the flood affected counties.” Seems trivial to me. If we select samples with value V>=1, we will always get an average greater than one. For example, the average number of times that people are struck by lightning is greater than 1 if we only count people who were struck by lightning at least once. As the authors are experts in statistics they are surely aware of this. Hence it is just a case of clarifying their meaning.

--4.2 1st Par. “This confirms the COPEWELL model’s predictions that CF prior to an event is a positive influence on sub-sequent CF trends.” I would say that this confirms that “prior events have a negative correlation with CF baseline”.  I do not see any implication for trends. Could the authors please explain

--4.4 2nd Par. “Another desirable measure of flood effects would by event.” What does his sentence mean?

Author Response

cc

Date:  October 11, 2022

To:      Climate Reviewers and Editors

From: Norma Kanarek, Ph.D.

Re:      Manuscript: climate- 1947954

Dear Reviewers and Editors,

On behalf of my co-authors, I reply to the thoughtful reviews from the journal climate of our manuscript “Flood-related federally declared disaster events and community functioning”. Attached in this document is a reply to each reviewer comment received. Additionally, I submit a revised manuscript with changes tracked and manuscript with changes accepted.

We very much appreciate the thoughtful reviews and think our response has improved the clarity of the manuscript.

Please let us know if there is any additional response needed to you.

Reviewer #1:

I have carefully read the book proposal by Kanarek et al. and find that it is suitable for publication after minor, yet extensive, revisions. This is a straightforward, though not minor, study. The authors apply a pre-existing model and conceptual framework (COPEWELL) linking an operationalization of societal well-being (“Community Functioning” or CF) to flood-related disasters. The questions of the study are simple and clearly outlined in the introduction. As the authors explain, these questions are not trivial given the mixed responses from the state-of-the-art literature which alternatively places some role or no role in community functioning of flooding events.

The authors answer these questions clearly in the conclusion section and in a more nuanced manner in the discussion section. In particular, the authors are attentive to some of the possible limitations of the dataset and the representativity. My comments and calls for revision are mostly connected to requests for clarification and apparent typos. My comments follow below.

 Comments

-The authors often use the term “precariousness”. Is this a deliberate decision not to use the term “vulnerability” which is used and defined in the IPCC? Is “precariousness” different? from “vulnerability”? Is the point irrelevant to this study?

We thank this reviewer for an astute observation of terminology. “Precariousness” was intentionally used to denote a non-jargon term for “the state of being uncertain or dependent on chance”. (defined by Oxford languages (via Google) definition #2). Precariousness is only part of what is included in a concept of vulnerability, however descriptive. The IPCCC Glossary shows, “Vulnerability is the degree to which a system is susceptible to, and unable to cope with, adverse effects of climate change, including climate variability and extremes. Vulnerability is a function of the character, magnitude, and rate of climate change and variation to which a system is exposed, its sensitivity, and its adaptive capacity.” (https://www.ipcc.ch/site/assets/uploads/2018/04/4.9a.html#v ) We stand by the use of this word to introduce pure locational risk, to examine risk as a continuum, and to expand our concern to economic issues.

 -(Intro, 1st Par, bottom line) Is it helpful to conflate economic disturbances with environmental disturbances? Are the authors suggesting that recovery from these events would be similar? I ask because, in the following paragraph, the only reference given is specific to economic disasters.

The cited reference was used 1) to initiate the broad conversation that is required when approaching “community functioning” as a concept (Links et al 2018). The Chang and Rose reference harkens to the multidimensionality of disasters, as does Links et al. in the definition of community functioning; 2) we, like Chang and Rose, also thought that economics are part of more traditionally environmental disasters. As an example, in our table 5, total earnings are positively correlated with baseline community functioning. We go on in paragraph 5 of the Introduction to give examples.  After consideration of the reviewer’s comment, we have added the phrase “affecting the economy” in paragraph 2 of the introduction.

-(Intro, 5th  Par) word “data” is missing, methinks.

We added data in paragraph three. This reviewer suggestion was helpful toward understanding the text.

-(2.1.2) Is “2010” an error?

Yes, 2010 was a initial modifier but to be consistent with elsewhere in the manuscript, we have added “2010” in the second paragraph of this section for clarity. Thank you.

-3.1 How are “typhoons” included here? Are these from the Pacific territories? Why would we include these? They are completely different than US-based communities. What is the percentage of Typhoon/(Typhoon + Hurricanes)?

We grouped flood events using the FEMA database’s terms, which are inclusive of coastal storm or tsunami, flood, hurricane or typhoon, and severe storms or tornados. (section 2.3.3) US counties are the units of analysis (2.3.1). Typhoons are neither separated from hurricanes nor enumerable. If any typhoon affects the US states, they are presumably very small in number.

-Discussion section:

--Could the authors please define “current” as in “current floods” and “floods ten years prior”? I understand that the most recent data is from 2015.

In section 2.3.1 We have renamed “current floods” as “study period” and specified the time intervals for prior five year period, prior ten year period and prior fifteen year period.

As the reviewer points out, we realize that there are many dates and many intervals so we have revised the definitions to be more textual than numeric and have tried to be consistent thereafter in the simplest language used.

--4.1, 1st Par, Last sentence. The sentence is not very clear to me. Unless I am mistaken the “)” should be moved back to include “and population change (2010-15)”. Either that or the sentence should be rewritten.

Thank you. Indeed, a close paren was missing.

--4.1 2nd Par. The statement “counties tended to have more than one flood event, as the average was one and a half events each for the flood affected counties.” Seems trivial to me. If we select samples with value V>=1, we will always get an average greater than one. For example, the average number of times that people are struck by lightning is greater than 1 if we only count people who were struck by lightning at least once. As the authors are experts in statistics they are surely aware of this. Hence it is just a case of clarifying their meaning.

We have clarified our intended message as follows: During this period the average was one and a half flood events among flood affected counties. Thanks for pointing out how the sentence could be improved.

--4.2 1st Par. “This confirms the COPEWELL model’s predictions that CF prior to an event is a positive influence on sub-sequent CF trends.” I would say that this confirms that “prior events have a negative correlation with CF baseline”.  I do not see any implication for trends. Could the authors please explain.

Thanks for requesting clarity. We had muddled the predictors of baseline CF and of CF trends. Both outcomes are predicted by prior floods. The sentence is now two sentences and reads as follows.

“This confirms that prior flood events have a negative correlation with baseline CF.  The COPEWELL model’s prediction that CF prior to an event is positively correlated with subsequent CF trends. “

--4.4 2nd Par. “Another desirable measure of flood effects would by event.” What does his sentence mean?

Thank you for pointing out that this sentence was incorrect. It now reads as follows. “Another desirable measure of flood effects would be CF change by each event, as for example CF change measured for Hurricane Sandy affected counties.”

Reviewer 2 Report

Dear authors,

Please recheck your paper. It seems you were in a rush to finish up your paper. Thus there are a number of typing omissions (unused superscripts in tables, missing interpunctions, etc.).

In addition, the superscripts like "*", "ns" etc. if not used, please delete, otherwise if needed, please indicate the meaning of these.

Otherwise, thank you for your contribution.

Author Response

Date:  October 11, 2022

To:      Climate Reviewers and Editors

From: Norma Kanarek, Ph.D.

Re:      Manuscript: climate- 1947954

Dear Reviewers and Editors,

On behalf of my co-authors, I reply to the thoughtful reviews from the journal climate of our manuscript “Flood-related federally declared disaster events and community functioning”. Attached in this document is a reply to each reviewer comment received. Additionally, I submit a revised manuscript with changes tracked and manuscript with changes accepted.

We very much appreciate the thoughtful reviews and think our response has improved the clarity of the manuscript.

Please let us know if there is any additional response needed to you.

Reviewer #2

Please recheck your paper. It seems you were in a rush to finish up your paper. Thus there are a number of typing omissions (unused superscripts in tables, missing interpunctions, etc.).

In addition, the superscripts like "*", "ns" etc. if not used, please delete, otherwise if needed, please indicate the meaning of these.

Otherwise, thank you for your contribution.

Thank you for the reminder to further proofread the text and tables. Minor changes are now shown in the manuscript with changes tracked.
